# Transcription Factors and Their Regulatory Roles in the Male Gametophyte Development of Flowering Plants

**DOI:** 10.3390/ijms25010566

**Published:** 2024-01-01

**Authors:** Zhihao Qian, Dexi Shi, Hongxia Zhang, Zhenzhen Li, Li Huang, Xiufeng Yan, Sue Lin

**Affiliations:** 1College of Life and Environmental Science, Wenzhou University, Wenzhou 325035, China; 21451335036@stu.wzu.edu.cn (Z.Q.); 21451335037@stu.wzu.edu.cn (D.S.); 22451335060@stu.wzu.edu.cn (H.Z.); 22451335019@stu.wzu.edu.cn (Z.L.); 2Laboratory of Cell & Molecular Biology, Institute of Vegetable Science, Zhejiang University, Hangzhou 310058, China; lihuang@zju.edu.cn; 3Zhejiang Provincial Key Laboratory for Water Environment and Marine Biological Resources Protection, Wenzhou University, Wenzhou 325035, China

**Keywords:** flowering plants, male gametophyte development, transcription factors, regulatory roles, downstream targets, upstream regulators

## Abstract

Male gametophyte development in plants relies on the functions of numerous genes, whose expression is regulated by transcription factors (TFs), non-coding RNAs, hormones, and diverse environmental stresses. Several excellent reviews are available that address the genes and enzymes associated with male gametophyte development, especially pollen wall formation. Growing evidence from genetic studies, transcriptome analysis, and gene-by-gene studies suggests that TFs coordinate with epigenetic machinery to regulate the expression of these genes and enzymes for the sequential male gametophyte development. However, very little summarization has been performed to comprehensively review their intricate regulatory roles and discuss their downstream targets and upstream regulators in this unique process. In the present review, we highlight the research progress on the regulatory roles of TF families in the male gametophyte development of flowering plants. The transcriptional regulation, epigenetic control, and other regulators of TFs involved in male gametophyte development are also addressed.

## 1. Introduction

Successful male gametophyte development is critical for plant reproduction, the creation of genetic diversity, and agricultural production [1]. Pollen development, pollen germination, and pollen tube growth, which are predominantly hidden within the tissues of the flower, are complex processes [2]. Angiosperm pollen ontogenesis is comprised of two sequential phases, a developmental phase, leading to the formation of mature pollen grains, and a functional phase, initiated right after the landing of pollen grains on the stigma and ending with double fertilization [3]. By gene-by-gene characterization, a considerable number of gametophytic/sporophytic tissue-expressed genes have been identified to be implicated in this extremely precise process of pollen ontogenesis [4,5]. It is estimated that about 14,000 genes and 25,000 transcripts are expressed in the male gametophytes of the dicot plant model organism *Arabidopsis thaliana* (hereafter, Arabidopsis) and monocot model rice (*Oryza sativa*), respectively [6,7]; however, the regulatory framework of the majority is still hiding somewhere outside our realm of cognition.

Transcription factors (TFs) in higher plants are proteins that interact with a specific DNA sequence and promote or repress the transcriptional activity of target genes [8]. They are typically composed of a DNA-binding domain, a transcription regulation domain, an oligomerization site, and a nuclear localization signal [7]. To date, the manipulation of more than 58 TF families in plant growth and development, as well as response to various environment stresses, has been demonstrated, including basic helix–loop–helix (bHLH) TFs, MYB TFs, and Lateral Organ Boundaries Domain/Asymmetric Leaves 2-like (LBD/ASL) proteins [9,10,11,12]. It is notable that in the last decade, a large number of TFs related to the male gametophyte development process, especially meiosis, microspore and tapetum development, and pollen wall formation, have been identified, such as MMD1, LBD27/10, DUO1, AMS, DYT1, and ARF17 [13,14,15]. A loss-of-function investigation of these TFs revealed considerable variations in morphological phenotype of anther/pollen behaviors, indicating the essential prerequisite of TF regulation for male gametophyte development. Furthermore, several TFs form regulatory cascades in determining the differentiation and development of anther and/or pollen [14,16,17,18,19].

Numerous studies have demonstrated the important biological functions of individual TFs in male gametophyte development [20,21,22,23]. Several excellent reviews have summarized genes and enzymes necessary for pollen development, especially the formation of the outer pollen wall named exine [1,15,24,25,26,27]. Recently, a review focusing on the molecular mechanism of TFs driving their functions in gametogenesis and sexual reproduction of non-seed plants and algae was available [28]. However, no relevant review exists to comprehensively sort out and summarize the research progress on the roles and their intricate coordinated regulation of various TFs during the male gametophyte development of flowering plants.

In this review, we summarize the current knowledge on the study of TFs associated with male gametophyte development in flowering plants, with emphasis on the regulatory roles of TFs in microspore development and tapetum function. In addition, we highlight recent advances in understanding the coordinated transcriptional regulation, epigenetic control, and other regulators of TFs involved in male gametophyte development.

## 2. Male Gametophyte Development of Flowering Plants

The acquisition of durable pollen grains, surrounded by an elaborately sculpted pollen wall capable of withstanding the harsh terrestrial environment, provides a guarantee for successful sexual reproduction and alternation of generation in flowering plants [29]. Pollen development is a highly conserved process stemming from anther cell division and differentiation, leading to male meiosis and germ cell formation, as well as pollen wall construction [30,31]. This complex process occurs inside the anther chamber, which is surrounded by an epidermis, an endothecium, a middle layer, and a tapetum from outside to inside [32]. In the plant model organism Arabidopsis, pollen mother cells (PMCs) derived from the archesporial cells generate a tetrad (Td) of four haploid spores surrounded by a callose wall following the first meiosis. Then, the callose is timely degraded by callase, which is produced by sporophytic tapetum, to dissolve the haploid microspore, which further undergoes an asymmetric mitosis, resulting in a generative cell and a vegetative cell. Subsequently, the generative cell undergoes further mitosis to form a tricellular pollen (TCP) with a vegetative cell and two sperm cells [33,34,35,36,37,38].

Along with the development of male gametophytes, the pollen wall is elaborately constructed simultaneously when individual microspores are released after callose degradation [15,25]. The fundamental structure of the pollen wall shows a significant similarity among different species, with an outer exine mainly composed of sporopollenin and an inner intine mainly consisting of pectin, cellulose, hemicellulose, and hydrolytic proteins [25,33,39]. The synthesis of the pollen wall starts at the Td stage when the precursors of sporopollenin secreted from the tapetum begin to deposit and assemble onto the primexine around the young haploid microspores. The basic exine structure is evident in uninucleate microspores (UNMs) and the mature exine structure is visually completed at the bicellular pollen (BCP) stage, with the outer sexine comprising tectum and radially directed bacula and the inner nexine. Simultaneously, the intine starts to develop at the UNM stage and constantly thickens by the BCP stage. During pollen maturation, tapetum remnants deposit as pollen coats (tryphine) and fill the cavities of sexine [13,25,40].

In the functional phase of male gametophyte development, pollen grains are adhered onto the stigma and activated by rehydration, triggering pollen germination [41]. Then, the pollen tube penetrates the stigma and delivers two sperm cells into the embryo sac for double fertilization [42].

## 3. Roles of TFs in Male Gametophyte Development

High-throughput technologies have enabled analysis of the pollen transcriptome on a global scale [6,7,43,44,45,46]. Although the transcriptome is highly reduced compared with sporophytic tissues, like roots and leaves, a large number of genes are active in male gametophytes, with a progressive decrease in the transcript diversity from UNMs to TCPs/germinated pollen grains (GPGs), indicating putative functions in male gametophyte development [4]. Tremendous efforts involving genetic and transcriptomic approaches demonstrated that hundreds of genes function in male gametophyte development [3,5,15,25,47]. In addition to nuclear genes, some mitochondrial genes were also involved in male gametophyte development and determining pollen fertility, with considerable variations in the morphological phenotype, particularly the microspore and tapetum behaviors arising from gene mutation [48,49,50]. How the expression of these genes is coordinated for the sequential male gametophyte development has not been well investigated. Analysis of transcriptomic data identified a set of TFs that are specifically or preferentially expressed during male gametophyte development [22,51,52,53]. In the last decades, a set of reverse genetic screens and forward genetic strategies have identified a batch of TFs with specific expressions and regulatory roles in male gametophyte development, demonstrating the undoubted implementation of TFs in the pollen ontogenesis of flowering plants. In the following, we highlight advances in the regulatory roles of TFs and their targets to obtain a deeper understanding of male gametophyte development (Figure 1).

### 3.1. bHLH TFs

#### 3.1.1. Structure and Classification of bHLH TFs

bHLH proteins are one of the largest TF families in plants [54]. They are characterized by a highly conserved bHLH domain, which comprises two functionally distinct regions: the basic region at the N-terminus with the highly conserved HER motif (His5-Glu9-Arg13) that determines the DNA binding activity and specificity, and the HLH region (two amphiphathic α-helices connected by a loop of variable length) required for the formation of homo- or heterodimers [54]. Based on evolutionary relationships, DNA-binding specificity, and the conserved amino acids and domains, bHLHs could be classed into six different groups, among which group B has the bulk of plant bHLHs, and twenty-six sub-groups [10,55]. The bHLH gene family expanded dramatically in higher plants, and there are approximately 162 and 111 bHLH genes in Arabidopsis and rice, respectively [54]. Several excellent reviews are available that address the importance of these TFs for the transcriptional regulation of genes that participate in many essential physiological and development processes, as well as environmental stress adaptation and tolerance in plants [56].

#### 3.1.2. Roles of bHLH TFs in Male Gametophyte Development

The sporophytic tapetum has been proposed to provide a cascade contribution to pollen development based on cytological and molecular investigation [57]. There are five conserved TFs that proved to be critical for tapetum fate determination, among which AMS and DYT1/AtbHLH022 are bHLH members. In Arabidopsis, the homozygous *ams* mutant showed abnormally enlarged tapetal cells, delayed callose degradation, and aborted microspores devoid of sporopollenin precursors [58,59]. In the tapetum, AMS as a master regulator has a dual role in pollen wall construction. It directly regulates an MYB TF MS188 for sexine formation and an AT-hook nuclear localized (ATL) family protein TEK, which further directly targets arabinogalactan protein (AGP)-encoding genes, such as *AGP6*, for nexine layer formation, respectively [13]. In addition, AMS directly targets 23 genes involved in tapetal development and pollen wall formation, including *ABCG26/WBC27* essential for sporopollenin precursor transport, *A6* involved in callose dissociation, *CYP703A2*, *CYP704B1*, and *KCSs* for very-long-chain fatty acid synthesis, *PKSA* and *TKPR1* for phenolic synthesis, and *EXLs* and *GRPs* for tryphine formation [58,59,60]. Rice *TDR*, an orthologue of *AMS*, has also been implicated in pollen wall development by regulating aliphatic metabolism, a mutation that exhibits degeneration retardation of the tapetum and middle layer and collapsed pollen without sporopollenin or pollen coat deposition [61,62]. DYT1/AtbHLH022 directly regulates the expression of an MYB gene *TDF1* and acts upstream of AMS, MS188, TEK, and MS1 for early tapetal development and pollen wall construction [63]. Although there is a high similarity between the downstream genes of *DYT1* and *AMS*, the identification of hundreds of *DYT1*- and *AMS*-specific genes indicated the specific functions of these regulators [64]. Furthermore, the relatively conserved roles of the homologies of *AMS* and/or *DYT1* in tapetal development have been also characterized in maize and tomato [65,66].

Disruption of rice *bHLH142/TIP2*, whose expression is restricted to anthers, caused pollen sterility by interfering with meiosis and tapetal programmed cell death (PCD) [23]. Moreover, the overexpression of *bHLH142* upregulated the expression of *EAT1/DTD1/bHLH141*, which positively regulated the expression of two aspartic protease genes, *AP37* and *AP25*, resulting in premature tapetal PCD [67,68]. All these findings emphasized the role of *bHLH142* as a central switch in early anther development. Three duplicated Arabidopsis bHLH genes, *bHLH089*, *bHLH091*, and *bHLH010*, together are important for anther development; the double and triple mutants of *bHLH089/091/010* progressively exhibited increasingly aberrant anther phenotypes with abnormal tapetum, delayed callose degeneration, and aborted pollen, whereas single mutants showed no discernible phenotypic alterations [22,69]. MYC2/bHLH6, MYC3/bHLH5, MYC4/bHLH4, and JAZ repressor-targeted MYC5/bHLH28 have redundant functions, and the *myc2*
*myc3*
*myc4*
*myc5* quadruple mutant exhibited severe defects in stamen development with delayed anther dehiscence and pollen maturation [70,71]. The function of *BONOBO1* (*BNB1*) and *BNB2*, two members of the bHLH VIIIa subfamily, in the asymmetric division during pollen development, has been also identified recently [72].

### 3.2. MYB TFs

#### 3.2.1. Structure and Classification of MYB TFs

As one of the most prevalent TFs in plants, MYB TFs have a modular structure with a highly conserved MYB domain at the N-terminus, which generally consists of up to four amino acid sequence repeats, each building three α-helices, and a variable transcription-activating/repressing domain usually located at the C-terminus [9]. According to the number of adjacent MYB repeats, plant MYB TFs are divided into 1R-MYB/MYB-related, R2R3-MYB, 3R-MYB, and 4R-MYB classes [9]. Advances in the functional investigation of MYB TFs since the identification of the first MYB gene *COLORED1* in maize emphasize their significance in a multitude of vital plant activities related to plant growth and development, including primary and secondary metabolism, plant tissue differentiation and development, stress responses, and especially male gametophyte development [73,74,75].

#### 3.2.2. Roles of MYB TFs in Male Gametophyte Development

To date, a total of 197 and 155 MYB TFs have been identified in the model plant Arabidopsis and rice, respectively [76]. Several MYB TFs, especially the largest subfamily R2R3-MYB TFs, have been implicated in tapetal function by genetic approaches. *TDF1/AtMYB35* and *MS188/AtMYB103/AtMYB80* encode two R2R3-MYB TFs. The male-sterile mutant *tdf1* exhibited severely impaired tapetal development and callose dissolution [77]. The knockout mutant in *OsTDF1*, the orthologue of Arabidopsis *TDF1*, exhibited a similar phenotype with *tdf1* with vacuolated and hypertrophic tapetal cells in rice [78]. Mutation in *MS188* interfered with tapetal development, callose dissolution, and exine formation, which negatively impacted microspore development and male fertility [79]. A recent loss-of-function study of *OsMS188* showed that mutant *osms188* displayed impaired tapetal degradation, an absence of sexine, and a defective anther cuticle [80]. Further study demonstrated that TDF1 might be associated with redox and cell degradation, while MS188 is involved in the biosynthesis of sporopollenin [64]. Two *GAMYB-Like* genes, *MYB33* and *MYB65*, have a redundant role in anther development in Arabidopsis, as neither of the single mutants *myb33* or *myb65* exhibited an overt phenotypic alteration, while the *myb33 myb65* double mutant displayed aberrant tapetum hypertrophy and premeiotic pollen abortion [81].

Previous studies showed that MYB108 and MYB24 are critical components of a jasmonate acid (JA)-mediated transcription cascade that acts downstream of MYB21 and regulates anther dehiscence and pollen maturation [82]. More recently, MYB21 and MYB24 were shown to act in a regulatory triad with MYB99, which regulates the phenylpropanoid biosynthesis for pollen coat patterning by controlling *TRANSKETOLASE2* expression [83]. Recently, another tapetum-expressed MYB2 has been found to directly activate the expression of protease genes *CEP1* and *βVPE*, which, in turn, regulate the tapetal PCD and pollen formation [84]. *CSA* encodes an R3R3-MYB TF that acts upstream of the monosaccharide transporter-encoding gene *MST8* and regulates sugar partitioning essential for pollen development in rice [85]. The silencing of *SIMYB33*, a GAMYB-like gene preferentially expressed in the pistils and stamens of tomatoes, caused delayed flowering, aberrant pollen viability, and decreased fertility, probably through modulating the sugar metabolism [86]. Another GAMYB homolog called *LoMYB33* is strongly expressed in pollen and anther at the late developmental stages of a lily. The overexpression of *LoMYB33* has been reported to cause adverse impacts on anther development and result in partial male sterility [87].

The study also shed light on the regulatory roles of MYB TFs in male germline development. The first male germline-specific R2R3-MYB TF DUO1/AtMYB125 was characterized and found to be essential for germ cell division and gamete specification during microspore development by activating a germline-specific regulon, including *MGH3*, *GEX2*, *GCS1*, and *CYCB1;1* [88]. Further investigation revealed that DUO1 acts upstream of two EAR motif-containing C2H2-type zinc finger proteins (ZFPs) DAZ1 and DAZ2, which interact with the corepressor TOPLESS and lead to transcriptional repression [14]. A recent study showed that another microspore-specific GAMYB AtMYB81 stimulates Arabidopsis pollen’s first mitosis. Mutant *myb81-1* pollen was arrested before pollen mitosis II and failed to establish two cell lineages essential for pollen development [89].

In addition to the robust roles described above, some MYB TFs are indispensable during the pollen functional phase. MYB109 was found to negatively modulate pollen tube growth by suppressing the pollen development regulator RABA4D in Arabidopsis [90]. *MYB97*, *MYB101*, and *MYB120* were pollen-expressed and redundant in the pollen tube reception of Arabidopsis, as single and double mutants exhibited no discernable defective phenotype, while the triple myb97 myb101 myb120 pollen tubes failed to stop growing in synergids, which resulted in drastically reduced fertility [91,92].

### 3.3. BRI-EMS-Suppressor 1 (BES1) Family Members

#### 3.3.1. Structure and Classification of BES1 Family Members

Brassinosteroids (BRs), plant-specific polyhydroxy steroidal hormones, regulate multiple processes during plant growth and development, including male fertility [93,94,95]. Brassinazole-Resistant 1 (BZR1) and BES1 are two key homologous TFs in BR signaling transduction, which, in turn, regulate thousands of target genes by binding to the E-box (CANNTG) or BR-response element (BRRE, CGTGT/CG) [96,97]. They belong to a family consisting of six members, including BZR1, BES1, BEH1, BEH2, BEH3, and BEH4 in Arabidopsis [96,98,99,100,101].

#### 3.3.2. Roles of BES1 Family Members in Male Gametophyte Development

Previously, BZR1, together with BES1-Interacting MYC-like proteins (BIMs), was found to bind *cis*-elements in the *Flowering Locus D* (*FLD*) promoter and the first intron of *FLC* to regulate flowering in Arabidopsis [102,103]. More recently, a quintuple mutant for *BES1*, *BZR1*, *BEH1*, *BEH3*, and *BEH4* was generated, showing impaired tapetum differentiation and microsporogenesis [101]. Further genetic and biochemical evidence demonstrated that BES1, which regulates BR-mediated gene expression, is activated by EMS1-TPD1-SERK1/2-mediated signaling to control tapetum and pollen development [101]. Among the diverse target genes of BES1, mutants for *SPL/NZZ*, *TDF1*, *AMS*, *MS1*, and *MS2* had reduced pollen production and pollen viability [104]. In addition, OSBZR1 in rice was found to directly promote the expression of *CARBON STARVED ANTHER* (CSA) which encodes an MYB TF, and CSA directly triggers the expression of sugar partitioning and metabolic genes to ultimately promote pollen development [105].

### 3.4. MCM1/Agamous/Deficiens/SRF (MADS) TFs

#### 3.4.1. Structure and Classification of MADS TFs

MADS TFs are widely found in eukaryotes and constitute a large gene family in plants. Currently, there are 107 genes encoding MADS TFs identified in Arabidopsis [106]. The defining feature of MADS TF family members is the presence of the MADS domain named for MCM1/Agamous/Deficiens/Serum Response Factor [107]. MADS TFs can be divided into two lineages, type I and type II, distinguished by exon–intron and domain structure, rates of evolution, developmental function, and degree of functional redundancy [108]. Type I MADS TFs are further subdivided into three groups, Mα, Mβ, and Mγ, based on their phylogeny and the presence of conserved motif at the C-terminus [108]; while type II, which are characterized by the presence of a distinct domain structure consisting of the MADS, intervening (I), keratin-like (K), and C-terminal (C) domains, are subdivided into MIKC^c^ and MIKC* sub-groups based on the number of the I domain-encoding exons and the differences in the K domain structure [109]. It has become clear that plant MADS TFs act throughout the whole lifecycle of the plants, including vegetative growth, pollen and embryo sac formation, and seed development [108,110,111,112]. In addition, MICK-type MADS TFs also play a role in plant responses to various biotic and abiotic stresses [113,114].

#### 3.4.2. Roles of MADS TFs in Male Gametophyte Development

MADS TFs are required to control the complex transcriptional networks regulating male gametophyte development. *AGL65*, *AGL66*, and *AGL104* are MIKC-type MADS-box genes in Arabidopsis. The loss of AGL65 protein significantly decreased pollen germination rates, while the double mutant of *AGL66/104* almost prevented pollen germination in vitro and affected pollen tube growth [115]. A triple mutant for *AGL65/66/104* had normal pollen morphology but displayed markedly reduced pollen competitiveness compared to WT [116]. In rice, *OsMADS62*, *OsMADS63*, and *OsMADS68* are preferentially expressed in mature pollen and have functional redundancy during late pollen development. Their triple knockout mutant showed a complete sterile phenotype with pollen that could not germinate [51]. The MIKC-type MADS-box gene *ZmMADS2* is pollen-expressed and essential for male gametophyte development in maize. The loss-of-function mutation in *ZmMADS2* generated through antisense technology resulted in arrested anther and pollen development [117]. The RNAi-mediated suppression of *SlGLO1*, a MADS-box gene highly expressed in tomato petals and stamens, caused severe male sterility and aberrant pollen [118].

### 3.5. WRKY TFs

#### 3.5.1. Structure and Classification of WRKY TFs

WRKY proteins make a large complex TF family in higher plants and comprise 74, 287, and 129 members in Arabidopsis, *Brassica napus*, and rice, respectively [119,120,121,122]. They contain a DNA binding domain of approximately sixty amino acids in length, characterized by one or two conserved WRKYGQK motifs at the N-terminus and a zinc finger-like motif formed by the conserved cysteines and histidines (C_2_H_2_-type: CX_4_CX_22-23_HX_1_H, C_2_HC-type: CX_7_CX_23_HX_1_C) [123,124]. WRKY proteins bind directly to the W box (TTGACC/T) DNA-binding site to repress or trigger the expression of their downstream targets. Based on phylogenetic analyses, the number of WRKY domains and the type of zinc finger-like motif, the WRKY proteins in flowering plants can be classified into three categories: Group I, Group II (which can be further classified into IIa, IIb, Iic, Iid, and Iie sub-groups), and Group III [119,125]. An increasing number of studies have demonstrated that WRKY TFs are involved in plant growth and development processes, as well as responses to biotic and abiotic stresses [126,127,128].

#### 3.5.2. Roles of WRKY TFs in Male Gametophyte Development

Some WRKY TFs are involved in male gametophyte development and abiotic stress responses during this process in flowering plants. Previously, the overexpression of pollen-specific *WRKY34* was shown to negatively affect the fertility of mature pollen in Arabidopsis [20]. More recently, WRKY34 was reported to function redundantly with WRKY2, and together they interact with VQ20 proteins to form complexes to modulate pollen function [129,130]. Triple mutants for these genes exhibited defects in pollen development, pollen germination, and pollen tube growth. Moreover, in Arabidopsis, the deletion of *WRKY2* and *WRKY34* resulted in a decreased expression of a target gene *GPT1* and a reduced accumulation of lipid bodies in pollen, ultimately leading to a decreased pollen germination rate and reduced pollen viability [131]. In addition, *WRKY34* expression is upregulated under cold stress, and mutation in *WRKY34* exhibited increased pollen viability after cold treatment. Further functional analysis indicated that *WRKY34* acts downstream of MIKC*-type MADS TFs and might be involved in the CBF signal cascade in mature pollen under cold stress [20]. *GhWRKY22* is mainly expressed during the late stages of cotton pollen and flower bud development, a mutation that caused defective pollen development with the dysregulation of genes involved in JA synthesis [132]. The overexpression of *WRKY27* in Arabidopsis caused abnormal anther dehiscence and decreased pollen viability, resulting in male sterility [133].

### 3.6. ZFPs

#### 3.6.1. Structure and Classification of ZFPs

ZFPs are among the most abundant proteins in plants. Numerous studies have revealed that ZFPs participate in the regulation of many developmental processes, hormone responses, and stress tolerance [18,134,135]. ZFPs are characterized by a zinc finger domain that forms a *ββα* configuration with a two-stranded antiparallel *β*-sheet and a short *α*-helix [136]. The binding of zinc stabilizes the folded finger-like polypeptide dimensional conformation so that it may facilitate interactions between the proteins and other macromolecules, such as DNA. Based on the number and position of cysteine and histidine residues that bind zinc ions, ZFPs can be divided into nine types: C_2_H_2_, C_8_, C_6_, C_3_HC_4_, C_2_HC, C_2_HC_5_, C_4_, C_4_HC_3_, and CCCH [136]. Of these, C_2_H_2_ ZFPs comprise the largest class and are most clearly characterized in plants. Currently, a total of 176, 189, and 118 C_2_H_2_ ZFPs have been identified in Arabidopsis, rice, and tobacco, respectively [137,138,139].

#### 3.6.2. Roles of ZFPs in Male Gametophyte Development

There is abundant evidence that ZFPs perform their functions in male gametophyte development through transcriptional or chromatin regulation. In petunias, seven ZFPs were found to be expressed sequentially during anther development, implying a regulatory cascade of these TFs [140]. Further investigation showed that the silencing of one of these ZFP genes, the anther-specific *MAZ1*, affected multiple aspects of meiosis, which included an inability of chromosomes to condense, a loss of meiotic synchrony and uncontrolled cytokinetic events, and pollen abortion [141]. Another ZFP gene, *TAZ1*, from petunias is tapetum-specific, the silencing of which caused premature degeneration of tapetum, defects in pollen wall formation, and extensive pollen abortion [142]. *BcMF20* was isolated from the flower buds of Chinese cabbage (*B. campestris*) and is highly similar to petunia *TAZ1*. It is specifically expressed in tapetum and pollen during the late developmental stages. The suppression of *BcMF20* expression resulted in the malformation of the pollen wall and finally caused pollen deformity and reduced germination rates [143]. *AtZAT4* encodes a C_2_H_2_ ZFP in Arabidopsis, and its T-DNA insertion mutant exhibited decreased silique length, seed setting, and pollen germination rates [144]. DAZ1 and DAZ2 are male germine-specific nuclear C_2_H_2_-type ZFPs. The double mutant *daz1 daz2* showed a class of bicellular pollen grains with a single germ cell-like nucleus, indicating that DAZ1 and DAZ2 are required for germ cell division and correct gamete differentiation [14].

Recently, the regulation of tandem CCCH ZFPs in anther/pollen development has also been highlighted. Arabidopsis C3H14 and its homolog C3H15 were demonstrated to redundantly regulate secondary wall formation and additionally function in anther development. The *c3h14 c3h15* double mutants produced few pollen grains. Subcellular localization and biochemical analyses suggested that C3H14 and C3H15 might function at both the transcriptional and post-transcriptional levels [145]. Another Arabidopsis CCCH ZFP gene, *AtC3H18*, is predominantly expressed in the developing microspores, and its gain-of-function mutant exhibited a male sterility phenotype. Further investigation suggested that *AtC3H18* may modulate pollen mRNA by regulating the assembly/disassembly of messenger ribonucleoprotein (mRNP) granules, thereby affecting pollen development [146]. CCCH ZFP genes *BcMF30a* and *BcMF30c* are substantially expressed during microgametogenesis and pollen germination in *B. campestris*. Both loss-of-function and gain-of-function mutants in *BcMF30a* and *BcMF30c* displayed aberrant pollen development [147,148]. Rice DCM1 protein contains five tandem CCCH motifs and interacts with nuclear poly(A) binding proteins in nuclear speckles. It is required for male meiotic cytokinesis by preserving callose from premature dissolution [149]. In Arabidopsis, the mutation of *CDM1*, a gene encoding a CCCH ZFP, also affected the expression of some callose-related genes [150].

### 3.7. LBD Proteins

#### 3.7.1. Structure and Classification of LBD Proteins

The *LBD* gene family encodes a class of plant-specific TFs that significantly impact plant growth and metabolism, particularly lateral branch and organ development [151,152]. Genome-wide analysis has identified a total of 42 and 31 *LBD* genes in Arabidopsis and rice, respectively [151,152,153]. LBD proteins comprise a conserved LOB domain at the N-terminus and a variable C-terminal region responsible for transcriptional activation/repression of target gene expression [11]. The LOB domain primarily consists of three parts: the C-block (CX2CX6CX3C) that binds to DNA, the leucine-zipper-like coiled-coil motif (LX6LX3LX6L) that dimerizes the proteins, and the GAS block (Gly-Ala-Ser). Based on the phylogenetic analysis and sequence similarities, LBD proteins can be divided into two categories: class I and class II. All class I LBD members contain a C-block, a GAS block, and a leucine zipper-like coiled-coil motif, and can be sub-grouped into four clades (IA, IB, IC, and ID), while class II members lack an intact leucine zipper-like motif and are sub-grouped into IIA and IIB [152,154].

#### 3.7.2. Roles of LBD Proteins in Male Gametophyte Development

Genetic approaches have revealed several LBD proteins that play critical roles during male gametophyte development. *Sidecar Pidecar/LBD27/ASL29*, which is dynamically expressed in microspore nuclei, is required for the proper timing and orientation of the asymmetric microspore mitosis [155]. Further investigation revealed that *LBD10* co-acts with *LBD27* to regulate male gametophyte development [155,156]. In addition to *LBD10* and *LBD27*, the functions of *LBD22*, *LBD25*, and *LBD36* in pollen development have been also identified. These five LBD genes exhibit spatially and temporally distinct and overlapping expression patterns and interact with each other to form heterodimers for their function in pollen development in Arabidopsis [157].

### 3.8. NAM/ATAF1/2/CUC1/2 (NAC) TFs

#### 3.8.1. Structure and Classification of NAC TFs

The NAC proteins, which constitute a large and widespread plant-specific TF family, have numerous functions, including plant development, senescence, cell wall biosynthesis, and abiotic and biotic stress responses [158,159,160,161,162]. Currently, a remarkable diversification of NAC genes has been addressed across the plant kingdom. For instance, 105 and 151 NAC members have been found in Arabidopsis and rice, respectively [163,164,165]. NAC proteins have a modular organization, consisting of a ~150 amino acid-conserved NAC domain at the N-terminus, which forms a seven-stranded antiparallel twisted β-sheet flanked by α-helices, and a variable C-terminal domain with transcriptional regulatory activity [166,167,168].

#### 3.8.2. Roles of NAC TFs in Male Gametophyte Development

The diverse roles of NAC TFs have been found in plants, especially their contribution to male gametophyte development. *ANAC092* is expressed in developing anthers in Arabidopsis, the overexpression of which suppressed pollen production and upregulated the expression of pollen development-associated genes, such as *DYT1* and *AMS* [169]. NST1 and NST2 are two NAC TFs that are redundant in regulating secondary wall thickening in anther walls and are required for anther dehiscence [170]. Another Arabidopsis *NAC* gene, *TAPNAC*, has a *cis*-regulatory element in its promoter to direct its specific expression in the tapetum, suggesting a potential role in tapetum function [171]. *AIF* is a *NAC*-like gene involved in anther dehiscence through regulating the expression of JA biosynthesis genes [172]. In maize, *ZmNAC84* has been demonstrated to directly bind and repress the expression of *ZmRbohH*, thereby affecting pollen development [173]. RNAi transgenic rice plants for Os07g37920, an NAC gene that is homologous to wheat GPC-B1/2, had reduced pollen viability and failed anther dehiscence [174]. In addition, the heterologous expression of a cotton NAC gene *GhFSN5* in Arabidopsis negatively regulated secondary cell wall biosynthesis and anther development, leading to collapsed and nonviable pollen and severe sterility [175].

### 3.9. Other TFs

In addition to the above-mentioned TF family members, there are many other TFs that play a role in male gametophyte development. MS1 is a plant homeodomain (PHD)-finger TF directly targeted by MS188 and controls the expression of sporophytic pollen coat proteins (sPCPs) in Arabidopsis [176]. Mutant *ms1* pollen grains showed abnormal exine development and were devoid of tryphine [177,178]. *OsERF101* encodes an APETALA2/ethylene-responsive element-binding protein (AP2/EREBP), which is predominantly expressed in flowers, particularly in the tapetum and microspores in rice. It was discovered that during reproductive development, pollen fertility and drought tolerance were compromised in knockout mutant and RNAi lines, whereas they improved in the *OsERF101*-overexpressing plants [179]. *HsfA2a* and *HsfA1a* are two heat stress TF (Hsf)-encoding genes that are mutually activated during the heat stress response process in tomatoes (*Solanum lycopersicum* L.). Previously, the suppression of *HsfA2a* expression was shown to reduce pollen thermotolerance during meiosis and microsporogenesis and cause pollen sterility [180]. Recently, HsfA1a was reported to maintain pollen thermotolerance by enhancing antioxidant capacity and protein repair and degradation, ultimately improving pollen viability and fertility [181]. Other TFs and their target genes involved in male gametophyte development are represented in Table 1.

## 4. The Upstream Regulators of TFs Associated with Male Gametophyte Development

### 4.1. Transcriptional Regulatory Cascades

Considerable evidence suggests that TFs form regulatory networks to control male gametophyte development. In Arabidopsis and rice, bHLH TFs have been reported to directly regulate the precise expression pattern of MYB genes [197]. In lily, LoUDT1 interacts with LoAMS to play a role in tapetal development [198]. Arabidopsis MYB-CC family members, γMYB1 and γMYB2, can be directly assembled to the *cis*-acting element of the Phospholipase *A2-γ* promoter, and γMYB2 can interact with γMYB1 to enhance its activity [199].

Molecular and biochemical evidence has further shown that different TFs form complexes, together with feed-forward and feedback regulatory loops, to facilitate the expression of downstream targets and contribute to male gametophyte development. Arabidopsis DYT1 activates the expression of its downstream *bHLH010*, *bHLH089*, and *bHLH091*, which further feedback to enhance the nuclear localization of DYT1, and together they promote *MYB35* expression and anther development [200]. The germline-specific MYB protein DUO1 sets up and later responds to the DAZ1/DAZ2 nodes, which are C_2_H_2_-type ZFPs, to ensure germ cell division and correct gamete differentiation [14]. In addition, AMS has been shown to interact with bHLH089, bHLH091, and ATA20, implying the complex transcriptional regulatory networks of pollen development [60].

Moreover, several TFs form a genetic pathway that regulates male gametophyte development. Recently, the core genetic pathway of DYT1-TDF1-AMS-MS188/MYB103/MYB80-MS1, which consists of five key TFs, was highlighted in an excellent review that described its critical role for tapetum development and pollen wall formation in Arabidopsis [57]. A relatively conserved genetic pathway composed of the homologies of the five key TFs was also proposed in rice and maize, indicating its conservation between monocots and dicots [78]. Mutations in any TF gene in this conserved genetic pathway caused pollen abortion and ultimately male sterility. Coincidentally, a recent study on watermelon anther under cold stress showed that failed tapetal degeneration might also be attributed to the dysregulation of these sporophytic tissue-related TF gene expressions [201]. In addition, an AtTTP-miR160-ARF17-CalS5 pathway was proposed with a regulatory role in callose synthesis and pollen wall patterning, among which the CCCH ZFP AtTTP is involved in miR160 maturation during pollen development [202]. In rice anther tapetum and pollen development, the transactivation of bHLH142 is directly modulated by GAMYB at the early stage of meiosis but repressed by TDR at the young microspore stage [203]. In addition, bHLH142 also acts downstream of UDT1/bHLH164 and upstream of EAT1 in a GAMYB-independent pathway [203].

### 4.2. Epigenetic Machinery

Epigenetic machinery is defined as gene-regulating activities with heritable characteristics that occur without alterations in the base sequences, including DNA methylation and imprinting, histone modification, chromatin remodeling, and the regulation of non-coding RNAs (ncRNAs) [204,205,206]. Increasing evidence shows that epigenetic machinery is also involved in the regulation of TF activation. An extensive data survey through transcriptomic approaches revealed the abundance of pollen/anther-preferential ncRNAs in diverse species [207,208,209,210,211,212,213], suggesting their possible regulatory roles in male gametophyte development. Notably, several long ncRNAs (lncRNAs) and microRNAs (miRNAs) have been demonstrated to serve as upstream regulators of TFs to participate in this unique process (Table 2). For instance, the overexpression of lncRNA *osa-eTM160* in rice negatively regulated osa-miR160 to enhance *osa-ARF18* expression during early anther development, leading to reduced seed setting and seed size [214]. The suppression of miR156/157 by high-temperature stress altered the expression level of *Squamosa Promoter Binding Protein-Like* (*SPL*) genes and excessively activated the auxin signal, leading to male sterility and anther indehiscence [215].

In addition, the importance of phosphorylation of TFs in the regulation of male gametophyte development has also been highlighted. The loss of MPK3/MPK6 phosphorylation sites in WRKY34 was found to compromise the function of WRKY34 [130]. The phosphorylation of ZmNAC84 at Ser113 by ZmCCaMK is required for the repression of the *ZmRbohH* promoter activity during pollen development [173].

Furthermore, other epigenetic modifications, such as chromatin remodeling, are also involved in TF activation during male gametophyte development. As a floral repressor, the vernalization-mediated epigenetic repression of the MADS-box TF member FLC has been attributed to the accumulation of histone H3 lysine 27 trimethylation (H3K27me3) and H3K4me3 and a reduction in H3K36me3, triggered by several regulatory lncRNAs [219,220,221,245]. Dynamic DNA methylation has been also reported to trigger MYB activation in response to abiotic stress [75,246], although the involvement of DNA methylation of TFs in the gametophyte development lacks sufficient experimental support. Further exploring those epigenetic regulators and their corresponding target TFs will deepen the understanding of the entire regulatory network of male gametophyte development.

### 4.3. Other Regulators

Encouraging evidence shows that many other regulators are also involved in male gametophyte development. Phytohormones, such as JA, abscisic acid (ABA), gibberellin (GA), BRs, ethylene, and auxin, play vital roles in male gametophyte development [247,248,249,250]. In Arabidopsis, exogenous ethylene activates the *EIN2-EIN3*/EIL1 signaling pathway in the tapetal layer, resulting in defects in tapetal development and ultimately male sterility [251]. Auxin plays important roles not only in the later phases of anther development but also in the early anther morphogenesis [196]. ARF17 is essential for pollen wall patterning in Arabidopsis by modulating primexine formation partially by directly regulating the expression of CalS5 [194]. Five SHI/STY TFs acting as direct regulators of *YUCCA* auxin biosynthesis genes affect anther organ identity, tapetal PCD, anther dehiscence, pollen viability, and pollen dormancy [196]. In rice, GA modulates anther development via the transcriptional regulation of GAMYB, which targets downstream genes, such as CYP703A3. The knockout mutant for CYP703A3 and *gamyb* mutant generated aborted microspores surrounded by a defective exine [210]. In the presence of JA, JAZ degradation is induced, and heterodimers of MYB21 or MYB24 with MYC TFs regulate stamen development [56]. MYC2, MYC3, MYC4, and MYC5 interact with R2R3-MYB TFs MYB21 and MYB24 to from the bHLH-MYB complex and cooperatively regulate the JA-mediated stamen development and seed production [71]. Moreover, MYC2 regulates the transcription of JAV1 and JAM1/MYC2-LIKE1, together with JAM1 and JAM3, to negatively affect JA-mediated male fertility [252,253]. MYB108 and MYB24 act downstream of MYB21 in a transcriptional cascade and redundantly function in stamen and pollen maturation in response to JA [82]. A series of BR biosynthetic and signaling mutants showed reduced pollen production, viability, and release efficiency, with suppressed expression of many key genes required for anther and pollen development [104]. In addition, ABA-triggered ROS accumulation in rice developing anthers has been recently implicated in tapetal PCD induction and heat stress-induced pollen abortion [249].

## 5. Conclusions and Future Perspectives

Revealing the functions of male gametophyte development-related genes and the regulation relation among key genes in this process thus becomes the basis point for understanding plant sexual reproduction. This not only provides valuable insight regarding male fertility but also helps with obtaining high crop yield and improving reproductive efficiency with essential theoretical and applicable meaning. Male gametophyte development is a well-coordinated process governed by a complex regulatory network involving genetic and epigenetic machinery. In this review, we address the knowledge of the involvement of TFs in this unique and important process, particularly in microspore development, pollen wall formation, and tapetum function. Current efforts have seen a big leap in the understanding of male gametophyte development. Nevertheless, concerns about this field mainly focused on the dicot plant model organism Arabidopsis and monocot model rice. Future challenges include the exploration of more genes, enzymes, and regulatory factors in various plant species, further investigations on the coordinated transcriptional and post-transcriptional regulation of these elements, and the establishment of a more comprehensive gene regulatory network involved in male gametophyte development.

## Figures and Tables

**Figure 1 ijms-25-00566-f001:**
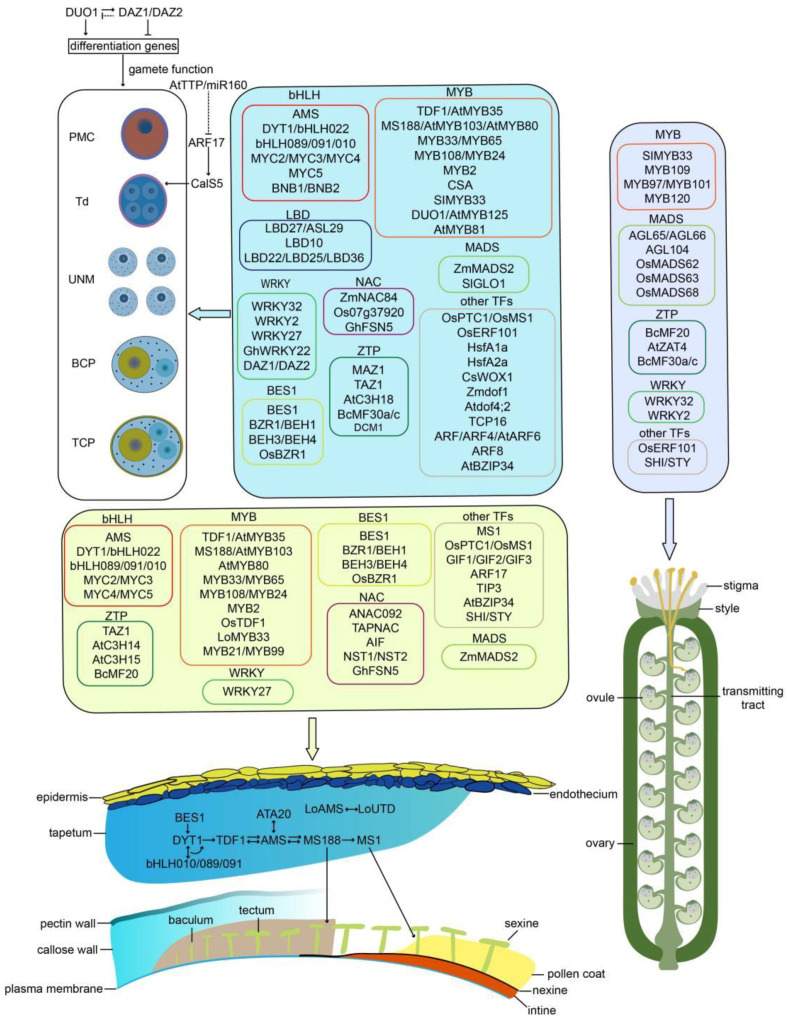
Roles of transcription factors in the male gametophyte development of flowering plants. BCP, bicellular pollen; PMC, pollen mother cell; TCP, tricellular pollen; Td, tetrad; UNM, uninucleate microspore.

**Table 1 ijms-25-00566-t001:** Other transcription factors and their target genes involved in male gametophyte development.

Transcription Factors (TFs)	TF Families	Target Genes	Species	Functions in Male Gametophyte Development	References
MS1	Plant homeodomain (PHD)-finger TF	/	*Arabidopsis thaliana*	Tapetal development and pollen wall formation	[176,177,178]
OsPTC1/OsMS1	PHD-finger TF	/	Rice (*Oryza sativa*)	Tapetal cell death and pollen development	[61,62,182]
OsERF101	APETALA2/ethylene-responsive element binding protein (AP2/EREBP)	/	Rice	Improving pollen fertility and seed sets under drought stress	[179]
HsfA1a	Heat stress TF (Hsf)	*Cu/Zn-SOD*, *GST8*, *MDAR1*, *HSP17.6A*, *HSP70-2*, *HSP90-2*, *HSP101*, *UBP5*, *UBP18*, *RPN10a*, and *ATG10*	Tomato (*Solanum lycopersicum* L.)	Pollen thermotolerance, pollen viability, and fertility	[181]
HsfA2a	Hsf	/	Tomato	Developmental activity and stress-regulated heat stress protection mechanisms in male gametophytic tissues	[180]
GmHSFA2	Hsf	*GmHSP20a*	Soybean (*Glycine max* (L.) Merr.)	Improving the heat tolerance during flowering	[183]
GIF1/GIF2/GIF3	GRF-Interacting Factors (GIFs)	/	*A. thaliana*	Anther development	[184]
CsWOX1	Wuschel-Related Homeobox (WOX)	*CsSPL*	*Cucumis sativus*	Early reproductive organ development, sporogenesis, and auxin signal transduction	[185]
Zmdof1	DNA-binding with one finger (Dof) protein	*Zm401*	Maize (*Zea mays* L.)	Pollen development	[186]
Atdof4;2	Dof protein	/	*A. thaliana*	Pollen development	[187]
TCP16	Teosinte Branched 1/Cycloidea/PCF (TCP) TF	/	*A. thaliana*	Early pollen development	[188]
TIP3	TDR Interacting rotein (TIP)	*TDR*	Rice	Formation of Ubisch bodies and pollen wall	[189]
ARF2	Auxin Response Factor (AFR)	/	*A. thaliana*	Floral organ abscission, leaf senescence, and flowering	[190,191,192]
ARF3/ARF4/AtARF6/ARF8	ARFs	/	*A. thaliana*	ARF3 and ARF4: floral organ development and male fertility; ARF6 and ARF8: floral maturation and hypocotyl development	[193]
ARF17	ARF	*MYB108*	*A. thaliana*	Pollen wall formation and tapetum development	[194]
AtbZIP34	Basic region/leucine zipper motif (bZIP) TF	*AtABCB9*	*A. thaliana*	Pollen development, pollen wall patterning, cell transport, and liposome metabolism	[195]
SHI/STY TFs (STY1, STY2, LRP1, SRS6, and SRS7)	Short Internodes/Stylish (SHI/STY) TFs	*EOD3*, *PAO5*, and *PGL1*	*A. thaliana*	Anther development and pollen germination	[196]

**Table 2 ijms-25-00566-t002:** Non-coding RNAs as regulators of transcription factors in male gametophyte development.

Non-Coding RNAs (ncRNAs)	Target Transcription Factors (TFs)	Species	Functions of Target TFs in Male Gametophyte Development	References
*Zm401*	ZmMADS2	Maize (*Zea mays* L.)	Microspore and tapetum development	[216]
*TaHTMAR*	TaBBX25 and TaOBF1	Wheat (*Triticum aestivum* L.)	Male fertility	[217]
lncRNA *osa-eTM160* as an endogenous repressor of osa-miR160	osa-ARF18	Rice (*Oryza sativa*)	Proper growth and organ development	[214]
lncRNA *bra-eTM160-1/2* as an endogenous target mimics (eTMs) *miR160*	BrARF17	*Brassica rapa*	Primexine formation and pollen development	[207]
*asHSFB2a*	HSFB2a	*Arabidopsis thaliana*	Both the female and male gametophytic development	[218]
*COLDAIR*, *COLDWRAP*, and *COOLAIR*	FLC	*A. thaliana*	Flowering	[219,220]
*MAS*	MAF4	*A. thaliana*	Flowering	[221]
*RIFLA*	OsMADS56	Rice	Flowering	[222]
*FLORE*	CDF5	*A. thaliana*	Photoperiodic flowering	[223]
miR156	SPLs	*A. thaliana*	Phase transition and flowering; anther development	[224]
	SPLs	Rice	Flowering	[225,226]
	NtSPLs	*Nicotiana tabacum*	Flowering	[227]
miR157	SPLs	Cotton (*Gossypium hirsutum*)	Pollen development and anther dehiscence	[215]
miR159	GAMYB-like TFs (MYB33/65/81/101/104)	*A. thaliana*	Anther development	[81,228]
miR159	OsGAMYB/OSGAMYBL1	Rice	Flower development	[229]
*TamiR159*	TaGAMYB1/2	Wheat (*Triticum aestivum*)	Heading time and male sterility	[230]
*Phe-MIR19*	PheMYB42/98	Moso bamboo (*Phyllostachys edulis*)	Anther dehisce, pollen separation, and seed formation	[231]
miR160	*ARF17*	*A. thaliana*	Callose synthesis and pollen wall patterning	[13,14]
miR167	*ARF6/8*	*A. thaliana*	Gynoecium and stamen/pollen development	[232,233,234]
*TaemiR167a*	TaARF8	Wheat	Male fertility	[235]
miR169	AtNF-YA TF	*A. thaliana*	Flowering	[236]
*zma-miR169o*	ZmNF-YA13	Maize	Seed development	[237]
miR171	GRAS family members	*A. thaliana*	Flowering	[238]
miR172	AP2	*A. thaliana*	Floral organ development and flowering	[224]
	GLOSSY15	Maize	Flowering	[239]
miR319a	TCPs	*A. thaliana*	Stamen development and anther dehiscence	[240,241]
miR396	GRF	*A. thaliana*	Anther development	[242]
miR824	AGL16	*A. thaliana*	Flowering in a long-day photoperiod	[243]
*TAS3* trans-acting siRNAs	ARFs	*A. thaliana*	Developmental timing and patterning	[244]

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
