# Peer review of "Transcription Factors and Their Regulatory Roles in the Male Gametophyte Development of Flowering Plants"

_ijms, 2024, doi:10.3390/ijms25010566_

Round 1

Reviewer 1 Report

Comments and Suggestions for Authors

The manuscript Transcription factors and their regulatory roles in male reproductive development of flowering plants by authors Zhihao Qian, Dexi Shi, Hongxia Zhang, Zhenzhen Li, Li Huang, Xiufeng Yan, Sue Lin presents a fairly thorough review of works on the role of transcription factors in the process of generative development plant spheres.

There are several problems with the terminology, general concept and availability of diagrams necessary to better understand and popularize this valuable and useful work. Also, I do not see any references to a number of general biological works that clearly give an idea of the botanical and anatomical aspects of the development of the generative sphere.

Terminology: male reproductive development - it is more correct to write the full name, otherwise it reads at least strangely (both in the title and throughout the text). I propose to replace it with “the development of the male gametophyte” or “the male gametophyte development”.

The connection between the individual parts of the manuscript could have been better if the authors had proposed a diagram or diagrams of the possible influence of TF on specific processes.

As for the references, it is not clear why the authors ignored a number of modern structural works in this area, which make it possible to see violations of the male gametophyte, for example, in transgenic plants. For example: Ji, J., Huang, J., Yang, L., Fang, Z., Zhang, Y., Zhuang, M., ... & Han, F. (2020). Advances in Research and Application of Male sterility in Brassica oleracea. Horticulturae, 6(4), 101; Chaban, I. A., Kononenko, N. V., Gulevich, A. A., Bogoutdinova, L. R., Khaliluev, M. R., & Baranova, E. N. (2020). Morphological features of the anther development in tomato plants with non-specific male sterility. Biology, 9(2), 32; Romani, F., & Moreno, J. E. (2021). Molecular mechanisms involved in functional macroevolution of plant transcription factors. New Phytologist, 230(4), 1345-1353 and others, where the analysis of morphology and genetics is clearly demonstrated.

Despite these shortcomings, if the problematic term is eliminated throughout the article, the work can be published.

Although, in my opinion, the diagrams and analysis of morphology should be extracted and expanded.

Reviewer 2 Report

Comments and Suggestions for Authors

The manuscript presented by Qian and colleagues is generally well structured and reviews clearly and linearly all the information related to transcription factors regulating the male reproductive development in flowering plants.

In my opinion, the manuscript is suitable for publication, however, I have some comments and questions which I include here below:

I would suggest rephrasing the first part of the abstract providing a small overview of the main elements that characterize the regulation of the male reproductive development in flowering plants.

The use of the term ‘epigenetic machineries’ is recurrent; it would be appropriate to describe what is meant by epigenetic machineries when first mentioned or in paragraph 4.2. Also I think it’s better to use the term ‘epigenetic machinery’ rather than machineries.

I would suggest standardizing paragraph headings e.g. ‘Structure and classification of…’ and ‘Roles of … in male reproductive development’.

Line 17: what is meant by ‘direct’ referred to analysis of transcriptome’, rephrase for clarity;

Line 18: change ‘suggests’ with ‘suggest’;

Line 23: add ‘the’ before ‘male reproductive development;

Line 31: ‘…short window of time’ I think it is appropriate indicate when these processes occur;

Line 37: change ‘trasncripts’ with ‘transcripts’;

Line 33: add ‘the’ before ‘stigma’;

Line 34: I suggest remove the expression ‘with convincing evidence’, it is superfluous;

Line 37: ‘…positive expression signal…’ means that 14,000 genes and 25,000 transcripts activate the gene expression, rephrase for clarity, maybe it would be more appropriate use the expression: ‘regulate the gene expression…’;

Line 45: change ‘a’ with ‘an’ before ‘oligomerization’;

Line 50: add ‘the’ before ‘male reproductive process’;

Line 57: remove s in ‘roles’;

Line 57-59: ‘the role of TFs …is receiving increasing attention…’ clarify the reason why the role of TFs in male reproductive development is receiving this increasing attention and clarify the importance of the male reproductive development for plant and production;

Line 58: ‘Various bodies of research’, rephrase for clarity;

Line 71: change ‘a’ with ‘an’ before ‘elaborately’;

Line 83: remove ‘a’ before ‘further’;

Line 85: add ‘the’ before pollen wall;

Line 86: remove ‘since’ and change ‘mircrospores’ with ‘microspores’;

Line 100: Figure 1A, indicate staining of the sections; verify the acronym ‘UMP stage’ and ‘BP stage’ in the figure, should be ‘UNM stage’ and ‘BCP stage’, also ‘MP stage’ is not defined;

Line 109: add the before embryo sac;

Line 125; change ‘implement’ with ‘implementation’;

Line 134: change ‘amphiphatic’ with ‘amphiphathic’;

Line 144: change ‘for’ with ‘to’ after ‘contribution’;

Line 145: remove ‘the’ before ‘cytological’;

Line 150: add ‘a’ before ‘dual’;

Line 159: remove ‘in’ before ‘which’;

Line 181: remove ‘s’ in ‘functions’ or change ‘has’ with ‘have’ before been also…;

Line 191: add ‘the’ before ‘functional investigation’;

Line 203: add ‘a’ before ‘phenotype’;

Line 215-216: change ‘act’ with ‘acts’ and ‘regulate’ with ‘regulates’ if the subject of the sentence is ‘jasmonate acid (JA)- mediated …cascade, remove ‘and’ before ‘more recently’;

Line 223: change ‘sliencing’ with ‘silencing’;

Line 230: add ‘the’ before study;

Line 241: add ‘the’ before ‘pollen functional..’;

Line 257: change ‘tapatum’ with ‘tapetum’;

Line 268: add ‘the’ before ‘expression’;

Line 274: change ‘domin’ with ‘domain’;

Line 283: change ‘became’ with ‘become’;

Line 309: add ‘s’ to ‘motif’ after WRKYGQK;

Line 334: remove ‘in’ before ‘which’;

Line 353: change ‘chromation’ with ‘chromatin’;

Line 367: change ‘specifc’ with ‘specific’;

Line 381: change ‘fuction’ with ‘function’

Line 384: remove ‘s’ in ‘Its’;

Line 385: change ‘Arbidopsis’ with ‘Arabidopsis’;

Line 405: change ‘specially’ with ‘especially’;

Line 433: change ‘biosynthsis’ with ‘biosynthesis’;

Line 450: change ‘was’ with ‘were’;

Line 470: change ‘showed’ with ‘shown’;

Line 459: Table 1 in ‘Function in male reproductive development’ I would recommend reducing the description e. g remove the expression ‘be required for’ or ‘plays an important role’ etc..;

Line 486: change ‘moncots’ with ‘monocots’;

Line 510: same comment as table 1;

Line 515: remove the doble ‘the’ before ZmRbohH;

Line 517: change ‘chromation’ with ‘chromatin’;

Line 528: change ‘may’ with ‘many’;

Line 552: change ‘have’ with ‘has’;

Line 560: add ‘a’ before ‘complex regulatory’;

Line 561: ‘the progresses on the..’ better a synonym such as insights or knowledge;

Line 566: add ‘the’ before ‘exploration’.

Comments on the Quality of English Language

I suggest a professional English editing service.
